# Deep Learning Versus Iterative Reconstruction for CT Pulmonary Angiography in the Emergency Setting: Improved Image Quality and Reduced Radiation Dose

**DOI:** 10.3390/diagnostics10080558

**Published:** 2020-08-04

**Authors:** Marc Lenfant, Olivier Chevallier, Pierre-Olivier Comby, Grégory Secco, Karim Haioun, Frédéric Ricolfi, Brivaël Lemogne, Romaric Loffroy

**Affiliations:** 1Department of Neuroradiology and Emergency Radiology, François-Mitterrand University Hospital, 14 Rue Paul Gaffarel, BP 77908, 21079 Dijon, France; marclenfant.ucbl@gmail.com (M.L.); pierre-olivier.comby@chu-dijon.fr (P.-O.C.); gregory.secco@gmail.com (G.S.); frederic.ricolfi@chu-dijon.fr (F.R.); brivael.lemogne@chu-dijon.fr (B.L.); 2Department of Cardiovascular and Interventional Radiology, ImViA Laboratory-EA 7535, François-Mitterrand University Hospital, 14 Rue Paul Gaffarel, BP 77908, 21079 Dijon, France; olivier.chevallier@chu-dijon.fr; 3Computed Tomography Division, Canon Medical Systems France, 24 Quai Gallieni, 92150 Suresnes, France; karim.haioun@eu.medical.canon

**Keywords:** computed tomography angiography, pulmonary embolism, artificial intelligence, image reconstruction, deep learning

## Abstract

To compare image quality and the radiation dose of computed tomography pulmonary angiography (CTPA) subjected to the first deep learning-based image reconstruction (DLR) (50%) algorithm, with images subjected to the hybrid-iterative reconstruction (IR) technique (50%). One hundred forty patients who underwent CTPA for suspected pulmonary embolism (PE) between 2018 and 2019 were retrospectively reviewed. Image quality was assessed quantitatively (image noise, signal-to-noise ratio (SNR), contrast-to-noise ratio (CNR)) and qualitatively (on a 5-point scale). Radiation dose parameters (CT dose index, CTDI_vol_; and dose-length product, DLP) were also recorded. Ninety-three patients were finally analyzed, 48 with hybrid-IR and 45 with DLR images. The image noise was significantly lower and the SNR (24.4 ± 5.9 vs. 20.7 ± 6.1) and CNR (21.8 ± 5.8 vs. 18.6 ± 6.0) were significantly higher on DLR than hybrid-IR images (*p* < 0.01). DLR images received a significantly higher score than hybrid-IR images for image quality, with both soft (4.4 ± 0.7 vs. 3.8 ± 0.8) and lung (4.1 ± 0.7 vs. 3.6 ± 0.9) filters (*p* < 0.01). No difference in diagnostic confidence level for PE between both techniques was found. CTDI_vol_ (4.8 ± 1.4 vs. 4.0 ± 1.2 mGy) and DLP (157.9 ± 44.9 vs. 130.8 ± 41.2 mGy∙cm) were lower on DLR than hybrid-IR images. DLR both significantly improved the image quality and reduced the radiation dose of CTPA examinations as compared to the hybrid-IR technique.

## 1. Introduction

Acute pulmonary embolism (PE) is the third most frequent cardiovascular disease, after acute myocardial infarction and stroke, causing approximately 37,000 deaths in Europe and 60,000–100,000 deaths in the USA each year [1,2]. Computed tomography pulmonary angiography (CTPA) is the first-choice diagnostic imaging modality for acute PE due to its wide availability and its minimal invasiveness [2,3,4]. In fact, approximately 2% of all emergency department patients undergo pulmonary CTPA for suspected PE [5]. However, the increased use of CT scans in emergency situations and the stochastic low-level radiation-related carcinogenesis raise concerns about long-term radiation exposure [6,7,8,9]. Although the radiology community applies the “As Low As Reasonably Achievable” principle for CT scans during scheduled exams, it is still important to demonstrate the efficacy and reliability of emergency low-dose CT scans [10,11]. Several approaches have been historically developed to reduce CT radiation dose, such as the use of fixed tube current reduction, automatic exposure control, and adjusted kilovoltage based on patient size and noise reduction filters [12]. While this has led to a significant dose reduction in many applications, image quality deterioration remained the main limitation for further radiation dose reduction, thus urging manufacturers to develop efficient reconstruction techniques.

Originally, CT images were reconstructed by filtered back projection (FBP). As the FBP algorithm assumes that the acquired projection data are free of noise, it requires low computational power, making it the method of choice for decades [13]. Nonetheless, when the radiation dose is too reduced or when large, heavy patients are examined, FBP produces noisy images that are susceptible to artifacts [14]. With the advancement of computational power, CT reconstruction techniques have undergone impressive development over the last 20 years, particularly iterative reconstruction (IR) techniques, which are now the new gold standard.

Briefly, IR can be divided into two main categories: hybrid-IR and model-based IR (MBIR) [15,16,17]. Even though hybrid-IR is the most widely used, its performance still has room for improvement compared to MBIR, which can achieve higher image quality and has advantages in terms of image noise, image texture, and spatial resolution. However, time-consuming reconstructions with MBIR is still an important issue, especially in the emergency setting. Recently, artificial intelligence has generated a high ground-swell of interest in several imaging applications, ranging from detection, recognition, and segmentation to a new type of reconstruction technique based on deep learning reconstruction (DLR) [18,19]. The first commercialized DLR tool, Advanced Intelligent Clear-IQ Engine (AiCE) (Canon Medical, Otawara, Japan), has been developed for CT and uses a deep convolutional neural network (DCNN) to distinguish true signal from noise within the image. For CT scan reconstructions, high-quality target data are acquired under optimized conditions and in the case of AiCE, are reconstructed with advanced MBIR. Image data that have been corrupted with artifacts and simulated noise that corresponds to 12.5–75% of the target image dose are then used for training (Figure 1) [20,21]. The DCNN algorithm promises high-quality reconstructions that bring the benefits of state-of-the-art advanced MBIR reconstruction image quality at much faster reconstruction speeds, increasing its applicability in the emergency CT workflow (Figure 2).

Previous DLR studies estimated that despite a 30% dose reduction, the quality of DLR images should be better than that of hybrid-IR images with reference dose [22,23]. However, the current version of AiCE requires 0.5 mm-thick native slices, resulting in slices twice as thin as in the imaging protocol performed with Adaptive Iterative Dose Reduction 3-Dimensional technique (AIDR 3D) from the same constructor and corresponding to hybrid-IR set up. On one hand, thinner slices should provide more accurate diagnosis in distal PE [24]; on the other hand, as the slice thickness is divided by a factor of two, it is expected to increase image noise by approximately a factor of 1.4 [25]. Based on these estimations, our initial experience in the clinical setting, and vendor recommendation, we intended to reduce the radiation dose by an estimated 20% in the AiCE group by increasing the noise index settings, expecting a significant dose reduction while still improving the overall image quality.

The aim of the present study was to investigate the impact of the DLR reconstruction technique (AiCE) on the image quality and radiation dose of CTPA compared to the current gold-standard hybrid-IR technique (AIDR) used as part of the PE imaging protocol in the emergency setting.

## 2. Materials and Methods

### 2.1. Study Population

A retrospective single-center study of all consecutive patients referred to our emergency department between November 2018 and January 2019 and who underwent a CTPA scan for suspected PE was employed. Patients were recruited using our Radiology Information System’s built-in thesaurus (XPLORE; EDL, La Seyne-Sur-Mer, France). For each search, a set of “PE” keywords (“pulmonary embolism”, “PE”) was combined with a “CTPA” filter using the Boolean operator “AND”. Exclusion criteria were as follows: allergy to iodinated contrast, severe renal disease, age < 18 years old, no recent (<6 months) body max index (BMI) assessment, and the use of a non-standard CTPA protocol. Patients with suboptimal enhancement of the pulmonary arterial trunk (<250 Hounsfield units) were secondarily excluded. Institutional Review Board approval and prior written informed patient consent were waived owing to the investigation’s retrospective nature of the study using existing CT images. Patients’ records and information were de-identified prior to analysis.

### 2.2. CT Scanning Protocol

All CT examinations were performed with a 320-detector row CT machine (Aquilion ONE GENESIS; Canon Medical Systems, Otawara, Japan). For each exam, the main scan parameters were as follows: helical acquisition, 0.5 mm × 80 rows, beam pitch of 0.813, 0.275 s gantry rotation time, 512 × 512 matrix, and 400 mm field of view. Automatic adjustment of the kVp was not available with the subtraction CT technique as all parameters (except the mA modulation) had to be identical for an accurate subtraction. Therefore, we decided to set up 100 kVp on our standard PE protocol in order to avoid beam hardening artifacts in patients with arms along the body or with a higher BMI. Automatic tube current modulation (^SURE^Exposure 3D) was used for all scans with a noise index set at 10.0 (defined as “Standard +1”) for AiCE with a mA range of 140–600 mA versus a noise index set at 8.5 with a mA range of 150–600 mA for AIDR 3D. The noise index settings (8.5 and 10) were based on our initial experience in the clinical setting and vendor recommendations. Reconstruction parameters were 0.5-mm slice thickness and 0.3-mm slice interval for the AiCE group and 1.0-mm slice thickness and 0.8-mm slice interval for the AIDR 3D group, respectively. Scan acquisition and reconstruction parameters are summarized in Table 1. In accordance with our standard CTPA protocol with the subtraction CT technique, all patients received a weight adapted 50–65 mL intravenous bolus of Iomeron 350 iodine per mL (Bracco Imaging, Courcouronnes, France), followed by a 50 mL chasing bolus of saline to guarantee sufficient contrast circulation in the pulmonary parenchyma and to obtain optimal iodine maps with the subtraction technique. Image acquisition was triggered using a predefined attenuation threshold in the pulmonary trunk at the level of the pulmonary artery bifurcation. As DLR reconstructions were available from December 2018 in our center, patients were allocated to the AIDR 3D group prior to this date and to AiCE afterwards.

### 2.3. Image Quality Assessment

#### 2.3.1. Objective Quantitative Evaluation of Image Quality

Images were blindly analyzed on a dedicated workstation by an independent radiologist with 3 years of experience in CTPA by manually placing circular regions of interest (ROI) within the axial images. He was different from the one who subjectively assessed the image quality. The ROI size was set as large as possible and positioned to avoid artifacts. Measurement of the image noise was performed by the standard deviation of the ROI in the descending aorta on soft filter images (Noise in the descending Aorta on Soft filter, NAS), in the trachea on soft filter images (Noise in the Trachea on Soft filter, NTS), and in the trachea on lung filter images (Noise in the Trachea on Lung filter, NTL). Signal-to-noise ratio (SNR) and contrast-to-noise ratio (CNR) were calculated using the method described by Szucs-Farkas et al., based on the following equation [26,27,28,29]:(1)$$\mathrm{SNR}=\frac{{\mathrm{SI}}_{\mathrm{vessel}}}{\mathrm{noise}}$$
(2)$$\mathrm{CNR}=\frac{{\mathrm{SI}}_{\mathrm{vessel}}-{\mathrm{SI}}_{\mathrm{muscle}}}{\mathrm{noise}}$$
where SI_vessel_ is the mean signal intensity (SI) of pulmonary vessels. SI_vessel_ was calculated as the average of the vascular enhancement measurements (in HU) obtained at five different levels (main pulmonary artery, right and left pulmonary arteries, right and left lower lobe arteries) and noise was defined as the mean of the standard deviation of these measurements. In the presence of an isolated PE, the ROI was placed in an adjacent vessel segment. SI_muscle_ was calculated as the average of the attenuation of the central part of the pectoral muscles and the deep paraspinal muscles, on both sides avoiding fat. The signal-to-noise ratio per unit dose (SNRD) and contrast-to-noise ratio per unit dose (CNRD) were then calculated, respectively, as follows [30]:(3)$$\mathrm{SNRD}=\frac{\mathrm{SNR}}{\surd {\mathrm{CTDI}}_{\mathrm{vol}}}$$
(4)$$\mathrm{CNRD}=\frac{\mathrm{CNR}}{\surd {\mathrm{CTDI}}_{\mathrm{vol}}}$$

#### 2.3.2. Subjective Qualitative Evaluation of Image Quality

All CTPAs were randomized, anonymized, and independently evaluated on standard LCD monitors by two radiologists with 3 and 7 years of experience, respectively, who were blinded to all patient data and reconstruction parameters. Readers were asked to subjectively rate the overall image quality on 0.5-mm (AiCE) and 1-mm (AIDR 3D) axial slices using a five-point scale ranging from 1 to 5: 1 = very poor image quality with no diagnostic information, 2 = low image quality that reduces the confidence in making diagnosis, 3 = moderate image quality sufficient to make diagnosis, 4 = good image quality clearly demonstrating anatomical structures, 5 = excellent image quality enabling excellent differentiation of even small anatomical structures. Observers also reported their diagnostic confidence to detect or exclude central, segmental, and subsegmental pulmonary embolic defects: 1 = no confidence; 2 = confident; 3 = very confident. Readers were able to adjust window level and width during the evaluation at their own discretion. To improve the interobserver agreement, the radiologists were blindly trained on images obtained from patients excluded from the study. In cases of disagreement (defined by 2 points or more of discordance), adjudication was performed by a third radiologist with 9 years of experience.

### 2.4. Radiation Dose Measurements

To assess radiation exposure, the volume CT dose index (CTDI_vol_) and the dose-length product (DLP) recorded as Digital Imaging and Communications in Medicine (DICOM) data were obtained.

### 2.5. Statistical Analysis

Continuous variables were presented as means ± standard deviation (SD) and dichotomous variables as numbers (percentages). The Kolmogorov–Smirnov test was used to test all parameters in both groups for normal distribution. As Gaussian distribution was present for all parameters except age, a two-sided paired *t*-test was used to evaluate differences between the hybrid-IR (AIDR 3D) and DLR (AiCE) groups regarding mean attenuation values of the DLP, CTDI_vol_, direct noise attenuation measurement, SNR, CNR, SNRD, CNRD, and image quality scores. A *p* value < 0.05 was considered as statistically significant. Interobserver agreement for qualitative analysis of image quality was evaluated by using Cohens’ kappa (*k*). Hereby, kappa values < 0 were considered as indicating no agreement, 0.00–0.20 as poor, 0.21–0.40 as fair, 0.41–0.60 as moderate, 0.61–0.80 as substantial, and 0.81–1.00 as excellent agreement. Subgroups analyses were also conducted according to the BMI (<25 kg/m^2^, 25–30 kg/m^2^, >30 kg/m^2^) in order to evaluate the impact of weight on image quality and radiation dose. All statistical tests were performed using the STATA^®^ software (version 13; StataCorp; College Station, TX, USA) and Excel Software (version 2016; Redmond, WA, USA).

## 3. Results

### 3.1. Patient Characteristics

One hundred and forty consecutive patients were retrospectively recruited. Ninety-three patients (49 men and 44 women) were finally included in the analysis, 48 in the AIDR 3D group and 45 in the AiCE group (Figure 3). The mean age of the patients was 67.9 ± 19.2 years (range, 19–92 years) at the time of AIDR 3D examinations and 69.0 ± 15.9 years (range, 26–91 years) at the time of AiCE examinations (*p* = 0.77). The mean BMI of the overall population was 24.4 ± 5.1 kg/m^2^ for the AIDR 3D group (range, 13.2–34.0 kg/m^2^) and 26.5 ± 6.2 kg/m^2^ (range, 17.0–40.6 kg/m^2^) for the AiCE group (*p* = 0.08). Patients were divided into three BMI categories for subgroup analysis: normal weight < 25 kg/m^2^ (*n* = 45), overweight 25–30 kg/m^2^ (*n* = 26), and obese > 30 kg/m^2^ (*n* = 22). There was no significant difference between BMI categories in terms of age.

### 3.2. CT Image Quality

#### 3.2.1. Quantitative Image Analysis

A total of 2.232 measurements were subjected to analysis (1.116 ROIs with mean signal and standard deviation). As shown in Table 2, the image noise was lower on DLR images whatever the ROI location, the filter used, and the BMI category. DLR-based reconstructions (AiCE) always showed a significant image noise decrease compared to the hybrid iterative-based reconstructions (AIDR 3D). In the overall population, mean decrease in image noise ranged from 27% to 32% on DLR images (Figure 4). Accordingly, SNR, CNR, SNRD, and CNRD were significantly higher on DLR images, increasing by 18%, 17%, 26%, and 26%, respectively. Across BMI categories, the image noise remained relatively stable on DLR images while steadily increasing on hybrid-IR images (Figure 5a). The linear regression analysis of SNRD according to BMI revealed a slower decrease of SNRD on DLR images, in particular for overweight and obese patients (Figure 5b).

#### 3.2.2. Qualitative Image Analysis

As shown in Table 3, in all patients, DLR yielded the highest overall image quality scores with both soft and lung filters (Figure 6). Subjective parameters were significantly improved in the global population, with a mean overall image quality score of 4.4 ± 0.7 and 4.1 ± 0.7 on DLR images with soft and lung filters, respectively, versus 3.8 ± 0.8 and 3.6 ± 0.9 on hybrid-IR images with soft and lung filters, respectively (*p* < 0.01). Overall image quality was better on DLR images in all BMI categories, but not significantly only for BMI > 30 kg/m^2^. There was no significant difference in diagnostic confidence level between the hybrid-IR and the DLR images. Inter-observer agreement was moderate with respect to the overall image quality (*k* = 0.59). Overall, 362 (77.8%) of 465 grades were the same between the two first observers, 90 (19.4%) differed by one point, and only 13 (2.8%) differed by two points, which led to an adjudication by the third radiologist. Ten of these 13 disagreements occurred in the DLR group, in obese patients (mean BMI = 30.8 ± 5.7 kg/m^2^), and the third radiologist chose the highest rating in 9 cases.

### 3.3. Radiation Exposure

The mean CTDI_vol_ values in the overall population for hybrid-IR (AIDR 3D) and DLR (AiCE) CTPA studies were 4.8 ± 1.4 mGy and 4.0 ± 1.2 mGy, respectively, corresponding to a significant radiation dose reduction of 17% with DLR compared to hybrid-IR (*p* < 0.01). Regarding the subgroup analysis, mean CTDI_vol_ values were significantly lower in the DLR group than in the hybrid-IR group whatever the BMI, <25, 25–30, or >30. The mean DLP values in the overall population for hybrid-IR (AIDR 3D) and DLR (AiCE) CTPA studies were 157.9 ± 44.9 mGy∙cm and 130.8 ± 41.2 mGy∙cm, respectively, corresponding to a significant radiation dose reduction of 17% with DLR compared to hybrid-IR (*p* < 0.01). Regarding the subgroup analysis, mean DLP values were significantly lower in the DLR group than in the hybrid-IR group for all BMI categories except >30. Data about radiation exposure are summarized in Table 4.

## 4. Discussion

The present study is the first to evaluate the effect of DLR on the image quality and radiation dose of CTPA in the emergency setting. DLR yielded a significantly lower image noise and higher SNRs and CNRs than hybrid-IR on both soft and lung images, whatever the BMI category. The subjective overall image quality score was significantly better with DLR than hybrid-IR for the evaluation of PE on both soft and lung images, even for overweight patients. We did not report any negative impact on our workflow using DLR images in the clinical routine as the AiCE reconstruction server allows parallel reconstruction during the acquisition. Currently, DLR reconstruction time is up to 40 images/s versus 3.5 image/s with MBIR technique (FIRST) and 80 images/s with hybrid-IR (AIDR 3D).

Interestingly, although the current version of AiCE requires thinner native slices than AIDR 3D (0.5 mm vs. 1 mm) and while the radiation dose was reduced by 17%, both quantitative and qualitative image quality were significantly improved. Indeed, image noise was reduced by approximately 25% on DLR images and SNR and CNR were increased by approximately 20%. Based on our findings, we suggest that as DLR allows thinner slices in CTPA images while improving image quality and reducing radiation exposure, DLR appears as an essential reconstruction method for CTPA scanning. These results are concordant with previous studies in other areas, which concluded that despite a radiation dose reduction of 30%, CT images reconstructed with DLR are still superior to the reference dose images reconstructed with hybrid-IR [18].

With the improvement of CT techniques, CTPA is now recognized as the technique of choice for diagnosis of suspected PE. However, the radiation dose has increased over time to compensate for the increasing image noise in thin-section imaging. Keeping the radiation exposure at minimum levels has become a major concern in the radiology community. Several dose-saving strategies have been proposed for reducing the radiation dose of CTPA, such as tube current modulation, lower kV, use of high-pitch protocol, application of shielding, and hybrid-IR technique [28,29,30]. DLR is currently the latest one, allowing one to reduce the radiation dose while improving overall image quality of CTPA, and could be considered as the new gold standard in the near future [31,32]. Indeed, some recent studies showed higher image quality of coronary CT with DLR versus hybrid-IR and higher image quality of abdominal ultra-high resolution CT with DLR versus MBIR [22,23]. The strength of the present study was that our sample of 93 patients was larger than prior studies using DLR. To our knowledge, this is the only and largest comparative study to date evaluating DLR versus hybrid-IR for CTPA.

Another important finding from the present study is the increased efficacy in image noise management reported for large and heavy patients with DLR. Indeed, even though SNRD and CNRD inevitably decrease in heavier patients due to the photo starvation, our study suggested a slower decrease with DLR than with hybrid-IR [33,34]. We hypothesize that the artificially decreased tube current in the training dataset mimics the photon starvation in large patients, thus helping to explain our findings. Additionally, our increasing experience with DLR (AiCE) has changed our approach to CTPA as most of the radiologists in our institution now use lung reconstructions for assessment of acute distal PE. Although soft filter images used to be ideal on hybrid-IRs for PE evaluation, the significant improvement in image noise reduction allowed by DLR combined with the greater spatial resolution offered by the lung filter seems to provide optimal results in terms of interpretation, without any significant trade-off as shown in Figure 6. Nonetheless, the potential benefits of 0.5 mm-thick slices in routine CTPA did not offset the fact that approximately 2000 images are produced for each CT exam, critically slowing down our Picture Archiving and Communication System (PACS). Fortunately, future improvements in AiCE should support thicker slice reconstructions that should be more appropriate in emergency settings, allowing a smoother workflow.

Our study has some limitations. First, the study population was relatively small and our investigation was retrospective and carried out at a single institution with two different but matched cohorts. Therefore, we consider our findings preliminary. Further studies on larger cohorts are underway to confirm our preliminary results. Second, diagnostic confidence levels were only evaluated on CT images with a soft filter for both AIDR 3D (hybrid-IR) and AiCE (DLR) reconstructions, which could have been less optimal for distal PE assessment. Third, we did not consider it relevant to assess the potential superiority of DLR over hybrid-IR to detect more PE, especially in subsegmental vessels, because of the low prevalence of PE in both groups. Instead, the goal of our study was to only evaluate the diagnostic confidence level of DLR, in addition to its impact on image quality and radiation dose. No difference was found for diagnostic confidence level between hybrid-IR and DLR. Lastly, our comparative study was performed between two reconstruction algorithms (AIDR 3D and AiCE) of the same manufacturer. Comparing different vendor-specific iterative reconstruction algorithms to DLR could be of interest. Additional investigations are needed.

## 5. Conclusions

In conclusion, DLR significantly improved image noise and the overall image quality of CTPA examinations in the emergency setting and offered an additional significant radiation dose reduction while allowing slices to be twice as thin as compared to hybrid-IR. Thus, DLR can yield better image quality than hybrid-IR while reducing radiation dose.

## Figures and Tables

**Figure 1 diagnostics-10-00558-f001:**
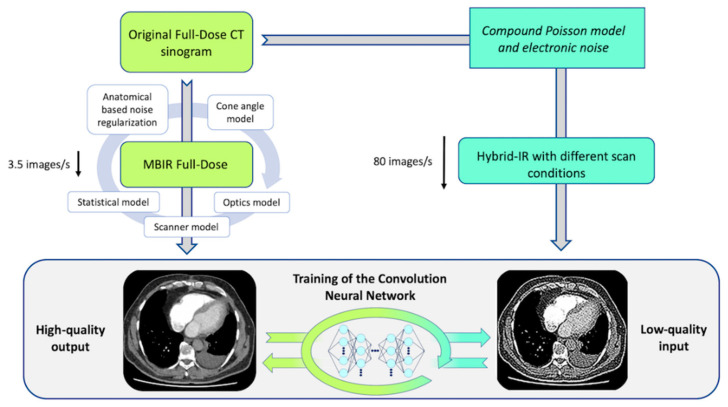
Schematic representation of the DLR training process. DLR, deep learning reconstruction; CT, computed tomography; IR, iterative reconstruction; MBIR, model-based iterative reconstruction.

**Figure 2 diagnostics-10-00558-f002:**
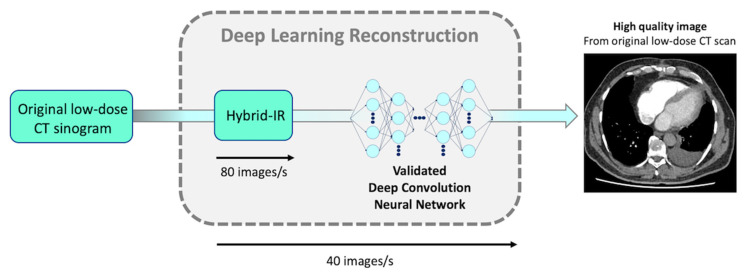
Schematic representation of DCNN direct application in clinical settings. DLR’s reconstruction time is up to 80 slices/s (similar to conventional hybrid-IR’s reconstruction time). DCNN, deep convolution neural network; DLR, deep learning reconstruction; CT, computed tomography; IR, iterative reconstruction.

**Figure 3 diagnostics-10-00558-f003:**
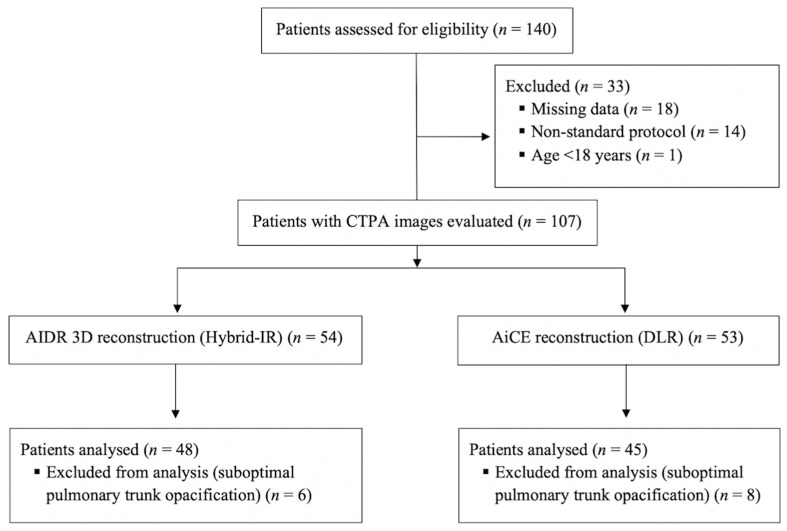
Flow chart of the study. CTPA, computed tomography pulmonary angiography, IR, iterative reconstruction; DLR, deep learning reconstruction; CT, computed tomography; AIDR 3D, adaptive iterative dose reduction three dimensional; AiCE, advanced intelligent clear-IQ engine; *n*, number.

**Figure 4 diagnostics-10-00558-f004:**
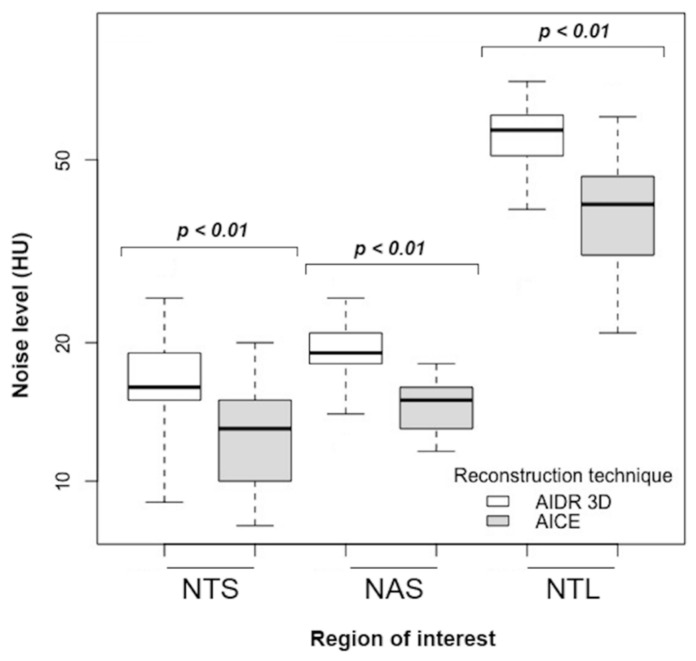
Image noise with hybrid-IR (AIDR 3D) and DLR (AiCE) studies in the overall population. Hybrid-IR, hybrid iterative reconstruction (AIDR 3D, adaptive iterative dose reduction 3-dimensional); DLR, deep learning reconstruction (AiCE, advanced intelligent clear-IQ engine); NAS, noise in the descending aorta on soft filter; NTS, noise in the trachea on soft filter; NTL, noise in the trachea on lung filter; HU, Hounsfield units.

**Figure 5 diagnostics-10-00558-f005:**
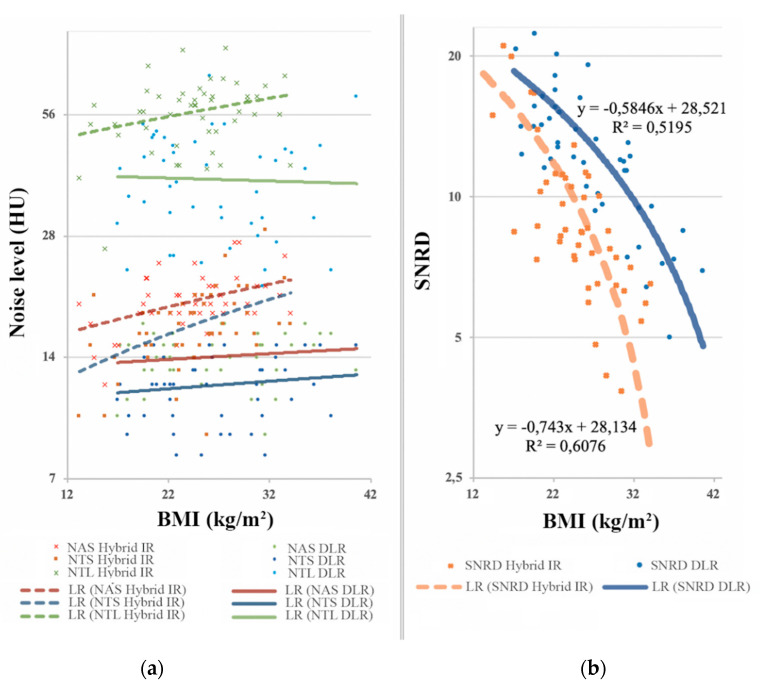
(**a**) Direct noise measurements based on BMI; (**b**) Evolution of signal-to-noise ratio per unit dose according to BMI. Across BMI categories, the image noise remained relatively stable on DLR images while steadily increasing on hybrid-IR images. The linear regression analysis of SNRD according to BMI revealed a slower decrease of SNRD on DLR images, in particular for overweight and obese patients. BMI, body mass index (kg/m^2^); HU, Hounsfield units; SNRD, signal-to-noise ratio per unit dose; NAS, noise in the descending aorta on soft filter; NTS, noise in the trachea on soft filter; NTL, noise in the trachea on lung filter; LR, linear regression; Hybrid-IR, hybrid iterative reconstruction (AIDR 3D, adaptive iterative dose reduction 3-dimensional); DLR, deep learning reconstruction (AiCE, advanced intelligent clear-IQ engine).

**Figure 6 diagnostics-10-00558-f006:**
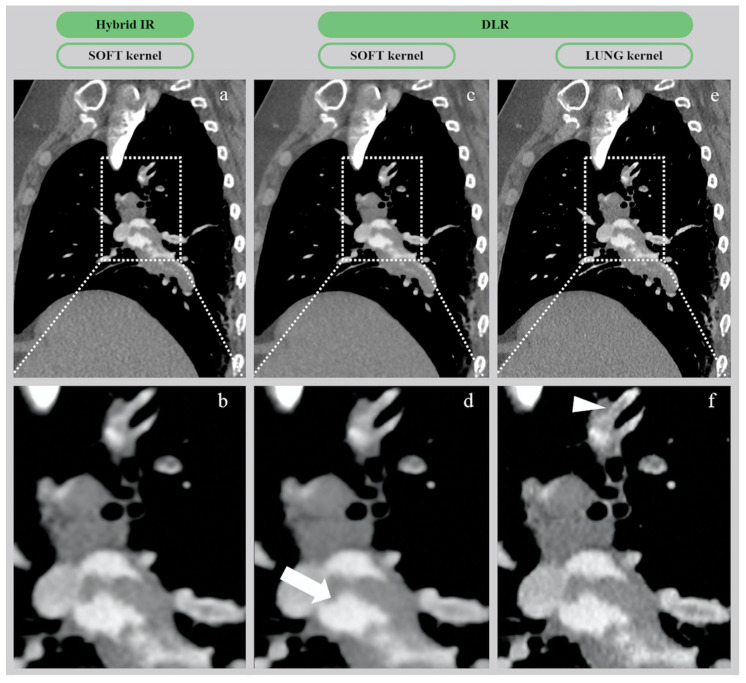
CTA images of pulmonary embolism in a 54-year-old woman (BMI = 42.5 kg/m^2^) with hybrid-IR (AIDR 3D) and DLR (AiCE) reconstructions. (**a**,**b**) Native and zoomed CT images from hybrid-IR (AIDR 3D) with 1 mm-thick slices on soft filter. (**c**,**d**) Native and zoomed CT images from DLR (AiCE) with 0.5 mm-thick slices on soft filter. (**e**,**f**) Native and zoomed CT images from DLR (AiCE) with 0.5 mm-thick slices on lung filter. On the zoom-in images, the DLR soft image appears as a virtually noise-free image (white arrow), whereas the DLR lung image provides the best spatial resolution, with excellent SNR/CNR depicting a clear subsegmental thrombus (white arrowhead). CTA, computed tomography angiography; Hybrid-IR, hybrid iterative reconstruction (AIDR 3D, adaptive iterative dose reduction 3-dimensional); DLR, deep learning reconstruction (AiCE, advanced intelligent clear-IQ engine); BMI, body mass index; CNR, contrast-to-noise ratio.

**Table 1 diagnostics-10-00558-t001:** CTPA acquisition and reconstruction parameters for hybrid-IR (AIDR 3D) and DLR (AiCE).

Parameters	AIDR 3D	AiCE
Reconstruction technique	Iterative	Deep learning
Acquisition mode	Helical	Helical
Tube voltage (kVp)	100	100
Tube current (mA)	150–500	140–600
Collimation (mm)	0.5 × 80 row	0.5 × 80 row
Rotation time (s)	0.275	0.275
Field of view (mm)	400	400
Slice thickness (mm)	1.0	0.5
Interval (mm)	0.8	0.3
Pitch	0.813	0.813
Noise index	8.5	10

CTPA, computed tomography pulmonary angiography; Hybrid-IR, hybrid iterative reconstruction; DLR, deep learning reconstruction; AIDR 3D, adaptive iterative dose reduction 3-dimensional; AiCE, advanced intelligent clear-IQ engine; kVp, kilovoltage peak; mA, milliampere; mm, millimeter; s, second; ms, millisecond.

**Table 2 diagnostics-10-00558-t002:** Quantitative analysis of CT image quality for hybrid-IR (AIDR 3D) and DLR (AiCE).

Parameters			AIDR	AiCE	% Change	*p*-Value
All patients, *n*			45	48	-	-
	Image noise (HU)					
		NAS	19.2 ± 3.0	14.0 ± 2.0	−27%	<0.01
		NTS	16.7 ± 3.7	11.9 ± 2.3	−29%	<0.01
		NTL	56.9 ± 10.8	38.7 ± 11.3	−32%	<0.01
	SNR		20.7 ± 6.1	24.4 ± 5.9	+18%	<0.01
	CNR		18.6 ± 6.0	21.8 ± 5.8	+17%	<0.01
	SNRD		10.3 ± 4.8	13.0 ± 5.0	+26%	<0.01
	CNRD		9.3 ± 4.6	11.7 ± 4.8	+26%	<0.01
BMI < 25, *n*			24	21	-	-
	Image noise (HU)					
		NAS	18.5 ± 2.9	13.9 ± 1.7	−25%	<0.01
		NTS	15.5 ± 2.9	11.9 ± 2.3	−23%	<0.01
		NTL	55.0 ± 11.5	39.6 ± 9.3	−28%	<0.01
	SNR		23.4 ± 6.2	27.1 ± 5.3	+16%	0.02
	CNR		21.3 ± 6.0	24.5 ± 5.4	+15%	0.05
	SNRD		12.7 ± 5.4	16.2 ± 5.0	+28%	<0.01
	CNRD		11.5 ± 5.1	14.6 ± 4.8	+27%	<0.01
BMI 25–30, *n*			18	8	-	-
	Image noise (HU)					
		NAS	20.1 ± 3.1	14.6 ± 1.9	−27%	<0.01
		NTS	17.8 ± 3.4	11.5 ± 2.9	−35%	<0.01
		NTL	58.7 ± 10.4	41.5 ± 16.1	−29%	0.02
	SNR		18.1 ± 4.3	24.6 ± 5.2	+36%	<0.01
	CNR		16.1 ± 4.5	21.9 ± 4.9	+36%	<0.01
	SNRD		7.8 ± 2.0	12.5 ± 3.5	+60%	<0.01
	CNRD		7.0 ± 2.0	11.1 ± 3.4	+59%	<0.01
BMI > 30, *n*			6	16	-	-
	Image noise (HU)					
		NAS	20.5 ± 2.6	14.0 ± 2.3	−32%	<0.01
		NTS	19.3 ± 5.3	12.3 ± 2.3	−36%	<0.01
		NTL	58.7 ± 9.9	36.1 ± 11.3	−39%	<0.01
	SNR		14.3 ± 2.0	20.8 ± 5.2	+45%	<0.01
	CNR		12.4 ± 1.8	18.2 ± 4.9	+47%	<0.01
	SNRD		5.8 ± 1.1	9.2 ± 2.5	+59%	<0.01
	CNRD		5.1 ± 1.0	8.1 ± 2.3	+59%	<0.01

Data are presented as mean ± standard deviation. *p*-values < 0.05 were considered significant. CT, computed tomography; Hybrid-IR, hybrid iterative reconstruction; DLR, deep learning reconstruction; AIDR 3D, adaptive iterative dose reduction 3-dimensional; AiCE, advanced intelligent clear-IQ engine; *n*, number of patients; %, percentage; BMI, body mass index (kg/m^2^); HU, Hounsfield units; NAS, noise in the descending aorta on soft filter; NTS, noise in the trachea on soft filter; NTL, noise in the trachea on lung filter; SNR, signal-to-noise ratio; CNR, contrast-to-noise ratio; SNRD, signal-to-noise ratio per unit dose; CNRD, contrast-to-noise ratio per unit dose.

**Table 3 diagnostics-10-00558-t003:** Qualitative analysis of CT image quality for hybrid-IR (AIDR 3D) and DLR (AiCE).

Parameters			AIDR	AiCE	*p*-Value
All patients, *n*			45	48	-
	Overall image quality				
		Soft	3.8 ± 0.8	4.4 ± 0.7	<0.01
		Lung	3.6 ± 0.9	4.1 ± 0.7	<0.01
	Diagnostic confidence level *				
		Proximal	3.0 ± 0.0	3.0 ± 0.0	-
		Segmental	2.9 ± 0.3	2.9 ± 0.3	0.93
		Subsegmental	2.6 ± 0.6	2.6 ± 0.7	0.90
BMI < 25, *n*			24	21	-
	Overall image quality				
		Soft	3.7 ± 0.9	4.4 ± 0.7	<0.01
		Lung	3.7 ± 0.9	4.2 ± 0.6	0.03
	Diagnostic confidence level *				
		Proximal	3.0 ± 0.0	3.0 ± 0.0	-
		Segmental	2.9 ± 0.3	3.0 ± 0.0	0.63
		Subsegmental	2.7 ± 0.5	2.8 ± 0.5	0.85
BMI 25–30, *n*			18	8	-
	Overall image quality				
		Soft	3.9 ± 0.6	4.6 ± 0.7	0.01
		Lung	3.5 ± 0.8	4.4 ± 0.5	0.01
	Diagnostic confidence level *				
		Proximal	3.0 ± 0.0	3.0 ± 0.0	-
		Segmental	2.9 ± 0.2	3.0	0.57
		Subsegmental	2.5 ± 0.7	2.9 ± 0.4	0.14
BMI > 30, *n*			5	16	-
	Overall image quality				
		Soft	4.0 ± 0.9	4.3 ± 0.9	0.6
		Lung	3.7 ± 1.0	3.8 ± 0.8	1
	Diagnostic confidence level *				
		Proximal	3.0 ± 0.0	3.0 ± 0.0	-
		Segmental	2.8 ± 0.4	2.8 ± 0.4	0.96
		Subsegmental	2.5 ± 0.8	2.3 ± 0.9	0.54

Data are presented as mean ± standard deviation. *p*-values < 0.05 were considered significant. CT, computed tomography; Hybrid-IR, hybrid iterative reconstruction; DLR, deep learning reconstruction; AIDR 3D, adaptive iterative dose reduction 3-dimensional; AiCE, advanced intelligent clear-IQ engine; *n*, number of patients; %, percentage; BMI, body mass index (kg/m^2^); * for evaluation of pulmonary embolism.

**Table 4 diagnostics-10-00558-t004:** Comparison of radiation dose between hybrid-IR (AIDR 3D) and DLR (AiCE).

Parameters		AIDR	AiCE	% Change	*p*-Value
All patients, *n*		45	48	-	-
	CTDI_vol_	4.8 ± 1.4	4.0 ± 1.2	−17%	<0.01
	DLP	157.9 ± 44.9	130.8 ± 41.2	−17%	<0.01
BMI < 25, *n*		24	21	-	-
	CTDI_vol_	3.9 ± 1.2	3.0 ± 0.8	−23%	0.02
	DLP	130.8 ± 37.9	100.7 ± 30.0	−23%	<0.001
BMI 25–30, *n*		18	8	-	-
	CTDI_vol_	5.4 ± 0.6	4.1 ± 0.8	−24%	<0.01
	DLP	179.6 ± 28.0	139.0 ± 34.4	−23%	<0.01
BMI > 30, *n*		5	16	-	-
	CTDI_vol_	6.2 ± 1.4	5.2 ± 0.6	−16%	0.04
	DLP	200.8 ± 47.2	166.3 ± 23.9	−17%	0.08

Data are presented as mean ± standard deviation. *p*-values < 0.05 were considered significant. Hybrid-IR, hybrid iterative reconstruction; DLR, deep learning reconstruction; AIDR 3D, adaptive iterative dose reduction 3-dimensional; AiCE, advanced intelligent clear-IQ engine; *n*, number of patients; %, percentage; CTDI_vol_, volume computed tomography dose index (mGy); DLP, dose-length product (mGy**∙**cm); BMI, body mass index (kg/m^2^).

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
