# Peer review of "Deep Learning Versus Iterative Reconstruction for CT Pulmonary Angiography in the Emergency Setting: Improved Image Quality and Reduced Radiation Dose"

_diagnostics, 2020, doi:10.3390/diagnostics10080558_

Round 1
Reviewer 1 Report
The aim of the study, which is to compare different reconstruction algorithms in CTPA, is important from clinical point of view.
The introduction sufficiently supports motivation of the study.
The aim is clearly stated.
The methods are well described and properly designed to achieve the aim.
The results are clearly presented with support of properly described and labeled Tables and Figures.
The discussion is well organized and includes all limitations of the study.
The conclusions are properly formulated and supported by the results.
Summarizing, the manuscript is well written and presents practically important findings. Minor language and style editing is required.
Author Response
Ref. Manuscript ID: diagnostics-885074 “Deep Learning Versus Iterative Reconstruction for CT Pulmonary Angiography in the Emergency Setting: Improved Image Quality and Reduced Radiation Dose”
Thank you very much for your valuable comments. Please find below a point-by-point reply to your comments. All changes have been marked in red (track changes) in the revised manuscript resubmitted to the journal.
Response to Reviewer #1 Comments
Point 1: The aim of the study, which is to compare different reconstruction algorithms in CTPA, is important from clinical point of view.
Response 1: Thank you very much for this comment. No additional correction has been made.
Point 2: The introduction sufficiently supports motivation of the study.
Response 2: Thank you very much for this comment. No additional correction has been made.
Point 3: The aim is clearly stated.
Response 3: Thank you very much for this comment. No additional correction has been made.
Point 4: The methods are well described and properly designed to achieve the aim.
Response 4: Thank you very much for this comment. No additional correction has been made.
Point 5: The results are clearly presented with support of properly described and labeled Tables and Figures.
Response 5: Thank you very much for this comment. No additional correction has been made.
Point 6: The discussion is well organized and includes all limitations of the study.
Response 6: Thank you very much for this comment. No additional correction has been made.
Point 7: The conclusions are properly formulated and supported by the results.
Response 7: Thank you very much for this comment. No additional correction has been made.
Point 8: Summarizing, the manuscript is well written and presents practically important findings. Minor language and style editing are required.
Response 8: Thank you very much for this comment. Minor language editing has been performed by a native speaker colleague, as requested.
Reviewer 2 Report
In this paper, the authors retrospectively compared the image quality and the radiation dose of computed tomography pulmonary angiography studies subjected to hybrid Iterative Reconstruction or to deep learning reconstruction. The work is interesting and I have just few comments:
1) All studies were performed with the same scanner (Aquilion ONE GENESIS, Canon Medical System) and the radiation exposure was assessed with CTDIvol and DLP.
Isn't it better to use a different parameter such as Size Specific Dose Estimate (SSDE) to assess patient dose reduction?
2) The Kolmogorov-Smirnov test was used to evaluate that all parameters, except age, followed normal distribution. Why didn't the authors use more powerful tests like the Shapiro-Wilk or Anderson-Darling test?
In table 2 change the first Imagenoise in Image noise:
Line 305 and 306 chane mGy.cm in mGy.cm
Author Response
Ref. Manuscript ID: diagnostics-885074 “Deep Learning Versus Iterative Reconstruction for CT Pulmonary Angiography in the Emergency Setting: Improved Image Quality and Reduced Radiation Dose”
Thank you very much for your valuable comments. Please find below a point-by-point reply to your comments. All changes have been marked in red (track changes) in the revised manuscript resubmitted to the journal.
Response to Reviewer #2 Comments
Point 1: In this paper, the authors retrospectively compared the image quality and the radiation dose of computed tomography pulmonary angiography studies subjected to hybrid Iterative Reconstruction or to deep learning reconstruction. The work is interesting and I have just few comments.
Response 1: Thank you very much for this comment. No additional correction has been made.
Point 2: All studies were performed with the same scanner (Aquilion ONE GENESIS, Canon Medical System) and the radiation exposure was assessed with CTDIvol and DLP. Isn't it better to use a different parameter such as Size Specific Dose Estimate (SSDE) to assess patient dose reduction?
Response 2: Thank you very much for this comment. The CTDIvol and DLP are the fundamental radiation dose metrics for CT to assess patient dose. Despite the lack of specificity regarding patient size that is included in the CTDIvol and DLP concept, these two metrics are the most common parameters used to evaluate dose reduction on the literature in CT. The SSDE seems to be interesting to assess patient dose reduction especially for pediatric patient. In fact, several studies have shown that the SSDE is a good surrogate for CT pediatric organ dosimetry, however, it has limits. Moreover, there’s now some software available which automatically calculate the SSDE by measuring the diameter on the patient either on CT localizer radiograph or on the axial images and after calculate the conversion automatically. Unfortunately, we do not have this software in our institution. Below are some examples of sources of uncertainties in SSDE estimates of patient organ dose, which explains why we preferred not to use the SSDE in our study:
- Patient Not Centered in the Gantry: some studies have shown that the SSDE, may be a poor predictor of organ dose when a patient is positioned either too high or low in the gantry, due to the impact of the bow-tie filter modulation on the x-ray beam, and over- or under-estimation of patient size using a CT localizer radiograph. For routine examinations, the SSDE may under- or overestimate organ dose by as much as 50% when a patient is not centered in the gantry.1 Another study has shown that the accuracy of the center slice SSDE approach correlated with patient size led to SSDE overestimation in small and underestimation in large patients.2 As the CT Pulmonary Angiography performed in our study have been scanned in the emergency condition, all patient could not be positioned with accuracy at the isocenter regarding the conditions of the patients.
- Partial Organ Irradiation: the SSDE organ dose calculation methodology assumes that the scan volume fully covers the organ, and thus partially irradiated organs, such as the liver during a chest CT scan, are not included in the overall chest radiation dose calculation. Therefore, the SSDE does not yield accurate estimates of dose for partially irradiated organs. The reasons cited above explain why we preferred not to use the SSDE for the evaluation of the dose reduction in our study. Therefore, we preferred to use the most common metrics used on the literature in CT: the CTDIvol and DLP.
- Zanca, F, Jacobs A, Crijns W, de Wever W. Comparison of measured and estimated maximum skin doses during CT fluoroscopy lung biopsies. Med. Phys. 2014;41:073901.
- Boos J, Kröpil P, Bethge OT, Aissa J, Schleich C, Sawicki LM, Heinzler N, Antoch G, Thomas C. Accuracy of size-specific dose estimate calculation from center slice in computed tomography. Radiat. Prot. Dosimetry. 2018;178:8-19.
Point 3: The Kolmogorov-Smirnov test was used to evaluate that all parameters, except age, followed normal distribution. Why didn't the authors use more powerful tests like the Shapiro-Wilk or Anderson-Darling test?
Response 3: Thank you very much for this comment. Our choice was based on previous benchmark studies in the field using the Kolmogorov-Smirnov test with similar numbers.1,2 However, we totally agree that many tests are possible with more or less statistical power. It does not seem absurd to use the Kolmogorov-Smirnov test in such a study. The Kolmogorov–Smirnov test can be modified to serve as a goodness of fit test. In the special case of testing for normality of the distribution, samples are standardized and compared with a standard normal distribution. This is equivalent to setting the mean and variance of the reference distribution equal to the sample estimates, and it is known that using these to define the specific reference distribution changes the null distribution of the test statistic. Various studies have found that, even in this corrected form, the test is less powerful for testing normality than the Shapiro-Wil test or Amderson-Darling test.3 However, these other tests have their own disadvantages. For instance, the Shapiro–Wilk test is known not to work well in samples with many identical values. Briefly stated, the Shapiro-Wilk test is a specific test for normality, whereas the method used by Kolmogorov-Smirnov test is more general, but less powerful (meaning it correctly rejects the null hypothesis of normality less often). However, which normality tests are "better" depends on which classes of alternatives we are interested in. One reason the Shapiro-Wilk is popular is that it tends to have good power under a broad range of useful alternatives.4 It comes up in many studies of power, and usually performs very well, but it's not universally best. Many people are using the Shapiro Wilk because it's often powerful, widely available and they are familiar with it but we cannot just use it under the illusion that it's "the best normality test". There isn't one best normality test. That’s why we decided to use the Kolmogorov-Smirnov test in the present study, without any relevant impact at the end compared to other tests given the sample. No additional correction has been made in the text.
- Yuan R, Shuman WP, Earls JP, Hague CJ, Mumtaz HA, Scott-Moncrieff A, Ellis JD, Mayo JR, Leipsic JA. Reduced iodine load at CT pulmonary angiography with dual-energy monochromatic imaging: Comparison with standard CT pulmonary angiography-A prospective randomized trial, Radiology. 2012;262:290–297.
- Fanous R, Kashani H, Jimenez L, Murphy G, Paul NS. Image quality and radiation dose of pulmonary CT angiography performed using 100 and 120 kVp. AJR Am J Roentgenol. 2012;199:990-996.
- D’Agostino RB, Stephens MA. Goodness of fit techniques, Marcel Dekker, New York, 1986.
- Chen L, Shapiro S. An Alternative test for normality based on normalized spacings.
Journal of Statistical Computation and Simulation.1995;53:269-287.
Point 4: In table 2 change the first Imagenoise in Image noise.
Response 4: Thank you very much for this comment. A suggested, imagenoise has been changed with image noise in table 2.
Point 5: Line 305 and 306 change mGy.cm in mGy.cm.
Response 5: Thank you very much for this comment. As requested, mGy.cm has been changed with mGy.cm through the manuscript in lines 305, 306 and 319.